# The Influence of Ultrasound-Guided Technique Using a Catheter-Over-Needle on the Incidence of Accidental Intravascular Injection during Caudal Block-A Retrospective Case Series Study

**DOI:** 10.3390/medicina57040318

**Published:** 2021-03-30

**Authors:** Daeseok Oh

**Affiliations:** Department of Anesthesia & Pain Medicine, Inje University Haeundae Paik Hospital, Busan KS012, Korea; yivangin@naver.com; Tel.: +82-517-970-415; Fax: +82-517-972-669

**Keywords:** caudal anesthesia, ultrasonography, catheters, contrast media

## Abstract

*Background and Objectives:* This study was conducted to investigate the influence of an ultrasound-guided technique using a catheter-over-needle when an intravascular injection occurs in patients undergoing a caudal block. *Material and Methods:* We retrospectively investigated 41 cases of an ultrasound-guided technique using an angiocatheter for caudal block. These had been performed between November 2019 through August 2020 to manage pain of lumbosacral origin. Under ultrasound guidance, after advancing the introducer needle through the sacrococcygeal ligament and then slowly withdrawing it, the outer catheter was continuously advanced into the sacral hiatus. We confirmed proper needle placement under fluoroscopic imaging, using 5 mL of contrast media. We assessed the contrast spread pattern with fluoroscopic imaging and checked it for the presence of intravascular injection. *Results:* In all recruited cases, the catheter-over-needle was guided successfully to the sacral hiatus and into the caudal epidural space. There was 100% accuracy under ultrasound guidance, without intravascular uptake, as confirmed by contrast media fluoroscopy. The incidence of ventral spread was 84.2% above the S1 with 5 mL of contrast. *Conclusions:* The technique of using a catheter-over-needle under ultrasound induced successful block without intravascular injection. Thus, this technique is a reliable option for conventional caudal block.

## 1. Introduction

Ultrasound (US) is excellent for guiding caudal epidural injections. It has similar treatment outcomes as does the fluoroscopically-guided technique, although it cannot provide information regarding injectate spread during caudal injection [1]. US has some advantages over fluoroscopy in guiding caudal blocks because it is easy to learn and radiation-free. Additionally, it can be used in virtually any clinical settings [2,3,4]. However, it has the fatal disadvantage of not being able to monitor for an inadvertent intravascular injection. This may lead to life-threatening complications or can result in a failed block [5,6]. Furthermore, a lack of blood flashback on pre-injection aspiration is not a reliable indicator of extravascular needle placement [7,8,9]. The occurrence of inadvertent vascular injection during caudal block with the conventional technique was 10.9–24.0% [8,10,11]. Previous studies of the conventional technique have noted intravascular injection occurring with a stiff metallic beveled needle.

We hypothesized that, if the flexible catheter is inserted through the sacral hiatus without touching bone, the incidence of intravascular injection might be minimized during the caudal block in accordance with the conventional technique. The aim of this study is to evaluate the influence of an ultrasound-guided novel technique using a catheter-over-needle on inadvertent intravascular injection occurrences in patients undergoing a caudal block.

## 2. Materials and Methods

### 2.1. Study Population and Data

This study was approved by the Institutional Ethics Committee of Inje University Haeundae Paik Hospital, Republic of Korea (HP IRB 2020-09-005). We retrospectively reviewed the electronic medical records, radiologic studies, anteroposterior (AP) and lateral fluoroscopic images acquired from a single pain clinic center during the intervention of 44 cases where US-guided caudal block with a catheter-over-needle had been performed between November 2019 and August 2020 because of lumbosacral origin pain. The etiology was analyzed, based on the patients’ medical records and review of imaging studies, including CT and MRI. Patients with a history of spinal surgery were excluded from the study.

### 2.2. Injection Technique

The US-guided caudal blocks were performed by a single pain physician with greater than 10 years of experience with caudal epidural injection under US guidance. We placed the patients in the prone position with the spine positioning system under the chest. A model LOGIQ^TM^ E10 7–14 MHz, multi-frequency linear US transducer was placed transversely at the midline to obtain a transverse view of sacral hiatus. We identified the hypoechoic region between the two band-like hyperechoic structures. At this level, we rotated the transducer to 90 degrees to obtain a longitudinal view of sacral hiatus. Under the longitudinal view, an 18-gauge, 1.88-inch, sterilized biosafety intravenous catheter (V4712-018-188, Sewoon Medical VINA Co., Ltd., Binh Duong, Vietnam) was inserted and guided under US to the sacral hiatus. After advancing the metallic guide needle through the dorsal sacrococcygeal ligament as a hyperechoic band-like structure in the US image and then slowly withdrawing it without contact with the bone, the outer catheter was introduced consecutively (Figure 1). We observed the hyperechoic guide needle’s advancement into the sacral hiatus in real-time imaging under the sonographic longitudinal view. We detected the characteristic “pop” as the sacrococcygeal ligament was penetrated. A hollow flexible tube was advanced until the green hub of the catheter reached to the skin surface. Rotating the outer catheter provides gentle insertion without kinking. We confirmed proper catheter placement with the fluoroscopy unit (OEC 9900 Elite, GE OEC Medical Systems, Salt Lake City, UT, USA) using 1.0 mL of contrast media (Omnipaque, 300 mg/mL, GE Healthcare, Little Chalfont, UK) after having confirmed negative aspiration. Optimal positioning was defined as a placement of catheter within the sacral epidural space on lateral view and an epidural filling pattern on AP view. After identifying the epidural space, we subsequently injected a total of 5 mL of contrast media to observe the spread pattern on AP and lateral views. A mixture of 13 mL of 0.2% ropivacaine, 5 mg dexamethasone and 1500 IU hyaluronidase (total volume 15 mL) were instilled for pain control. We saved all fluoroscopic images that we had done with contrast media.

### 2.3. Image and Data Analysis

Two pain physicians reviewed all of the images on a picture archiving and communication system. We checked the intravascular pattern, as well as the intravascular-plus-epidural pattern, on fluoroscopic view. The case was considered to be intravascular if fluoroscopic indicated a snake-like flow pattern or a vascular run-off with disappearance of the contrast. A mixed epidural and vascular pattern was defined as the contrast showing a transient fleeting pattern and epidural spread [12]. The spread level of the contrast media into the ventral epidural space was identified on lateral views. The final upper level was recorded. Radiologic criteria for each level followed the definition shown in a previous study [13].

### 2.4. Statistical Analysis

SPSS version 24.0 (IBM corporation, Armonk, NY, USA) was used to analyze the data. The data that are presented includes the frequency and percentage of categorical variables and a mean ± standard deviation (SD) for numeric variables.

## 3. Results

Among 44 cases whose data were collected, we excluded three cases from our analysis for the following reasons: an angiocatheter had not been used (*n* = 2); fluoroscopic lateral view image was not saved (*n* = 1) (Figure 2). A total of 41 cases that had been performed on 40 patients (one patient had two cases) was used in this study. There was a total of 20 males and 21 females. The average age was 59.1 ± 16.1 years (range, 21 to 82 years). Fluoroscopic findings confirmed that there was 100% accuracy in catheter placement into the caudal epidural space under US guidance without intravascular uptake. All catheter tips were located at the S3 vertebra, and all cases showed cephalic epidural spread of contrast (Figure 3). No intrathecal spread was found all cases. Adverse events during the procedure were not reported. The clinical data are presented in Table 1 and Table 2.

## 4. Discussion

In our 41 cases, there was 100% accuracy of the US guidance in locating the sacral hiatus for proper placement of the outer catheter into the caudal epidural space. No intravascular patterns were observed. This technique appears to be easy to learn and may be more comfortable for the patient.

The US-guided caudal block was first described in 2003 [14]. The authors suggested that a real-time sonographic method may be a good alternative to the current standard of fluoroscopic guidance. Thereafter, there was 100% accuracy in the caudal epidural needle placement into the caudal epidural space under US guidance [4]. However, the disadvantage is that the US cannot check for intravascular injection as compared to fluoroscopic guidance [5,6]. This is important because aspiration or return of blood is neither sensitive nor specific for the intravascular positioning of the needle [7,8,9]. This vascular injection vulnerability during caudal block using a conventional technique had already been well known. Caudal routes showed a 10.9% (128/1211) occurrence of intravascular uptake during fluoroscopically guided lumbar spinal injection procedures [8]. Fluoroscopic injection can check for vascular uptake with contrast in the process, whereas with US, inadvertent intravascular injection cannot be identified.

There are several factors that affect intravascular injection in caudal blocks, although it is unclear yet why the caudal route produces such a high incidence of intravascular injections. First, the epidural anatomy varies among lower back pain patients [15]. Second, impaired systemic venous return with aging can also lead to an engorgement of the epidural venous plexus [12]. The presence of radicular symptoms in the lumbar spine and the duration of those symptoms are significant and are independent risk factors for an accidental intravascular injection during caudal block, because it leads to a localized neovascularization [16]. There were no intravascular uptake findings, although our study included a substantial number of chronic radicular pain cases. Third, the epidural venous plexus is gathered at the anterior part of the sacral canal and ends at the S4 level or lower [17]. In the conventional technique, the needle comes into contact with the anterior wall of the sacral spine because of the kyphotic nature of the sacrum [10,11]. Doo et al. [11] reported that intravascular injection incidences during a US-guided conventional technique with a disposable nerve blockade needle (25-gauge, 5 cm short-bevel needle) was 24.0% (6 of 25), as confirmed by fluoroscopy. Park et al. [10] reported that intravascular injection occurrences were 20.3% (13 of 64) during a fluoroscopically guided conventional technique using a spinal needle (22-gauge, 8 cm Quincke) advanced to the mid-S3 level. They mention that unintentional intravascular injection might be caused by needle trauma in the vessel-rich zone in the sacral canal and direct needle contact to the sacral bone during the execution of the conventional technique. Even in some cases, the needle penetrated the posterior surface of the sacrum base after passing through the sacrococcygeal ligament [18]. We suggest that using a catheter-over-needle instead of the conventional method might reduce the possibility of touching the vascular structures. The advantage of a soft epidural catheter on intravascular injection occurrence during lumbar epidural block has been reported [19].

There might be a few explanations for our results. First, the introduced outer catheter characteristics could have contributed to reducing the inadvertent intravascular placement into the sacral canal. The outer catheter is made of polyurethane, which is an elastic material. The catheter tip is of a relatively small size and is rounded, without a bevel. Second, the US guidance technique can enable the puncturing of the sacrococcygeal ligament more accurately without touching the bone margin with the stylet. This can minimize tissue injury along the needle trajectory. The catheter-over-needle US-guided caudal block method is somewhat similar to the vascular catheterization technique. The sacrococcygeal ligament and caudal epidural space correspond to the vessel wall and the intravascular space, respectively. Therefore, if the outer catheter is introduced and a metallic guide needle is removed before it has completely entered the caudal epidural space, the outer catheter may not advance or it may kink. Furthermore, as outer catheter resistance could be affected by its outside diameter, wall thickness and the elastic properties of its material [20], different types of angiocatheter might need to be considered. We successfully placed an 18-gauge, 1.88-inch catheter into the caudal epidural space.

In our cases, we identified an 84.2% ventral spread occurrence with 5 mL of contrast above the S1. Doo et al. [11] have shown that the incidence of inadvertent intravascular injection is 0%, and the rate of successful procedure is 96% during the alternative method, which involves performing the injection right after penetrating the sacrococcygeal ligament, under US guidance. However, the detailed contrast spread pattern and analgesic efficacy of the alternative technique were not evaluated. The advantage of our technique compared to the alternative method is to advance the flexible catheter beyond the sacrococcygeal ligament. Therapeutically, we delivered a caudal block injectate for lumbosacral pain to above S1 level. The final level of needle tip could affect contrast distribution and analgesic effects. Additionally, Park et al. [10] reported that they saw a ventral epidural spread above the S1 in 94.1% of patients when they had used 15 mL of contrast through a 22-gauge spinal needle advanced to the mid-S3 level. The needle gauge did not seem to affect the level of epidural spread in the caudal blocks [13].

This study has several limitations. First, this study was designed as a retrospective chart review. It might be impossible to completely detect a vascular spreading pattern, because these patterns could be ambiguous despite the use of real-time fluoroscopy and manual blood aspiration. Further prospective studies using imaging modalities such as continuous fluoroscopy mode and digital distraction angiography should be done to clarify this. In addition, we did not evaluate clinical outcomes. The main limitation of this study was that it was a single study. It did not compare the effectiveness of our technique to that of other techniques. However, its effectiveness appears to be better compared to the published results of other methods. We believe that US is more sensitive than fluoroscopy for checking the sacrococcygeal ligament, although intravascular occurrences of fluoroscopic-guided techniques using an angiocatheter have not been reported yet. Despite the above limitations, this is the first report on the occurrence of inadvertent vascular injection during a US-guided technique with an angiocatheter for conventional caudal block. Further controlled studies with a large sample size are needed to determine advantages of this caudal block method to minimize the risk of intravascular uptake.

## 5. Conclusions

US-guided caudal block using a catheter-over-needle can avoid inadvertent intravascular injection that has been a problem during caudal block in accordance with the conventional technique. This technique should be preferred in clinical practice for safety reasons.

## Figures and Tables

**Figure 1 medicina-57-00318-f001:**
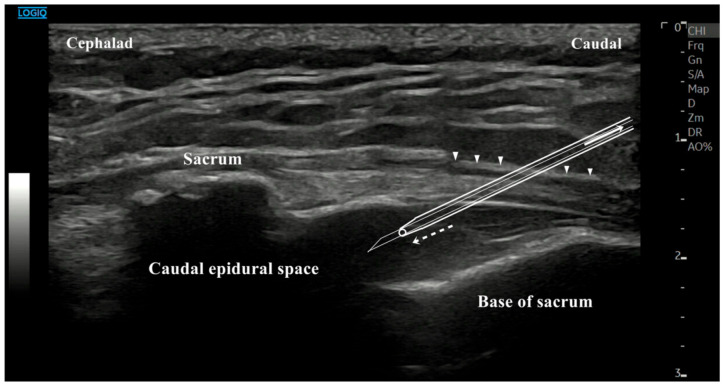
Sonographic image and schematic drawing of the ultrasound (US)-guided technique with an angiocatheter for caudal block. The outer catheter is advanced beyond the sacrococcygeal ligament (white arrow heads) into the caudal epidural space (dotted white arrow). The metallic guide needle should be withdrawn without making contact with the sacral bone (white arrow).

**Figure 2 medicina-57-00318-f002:**
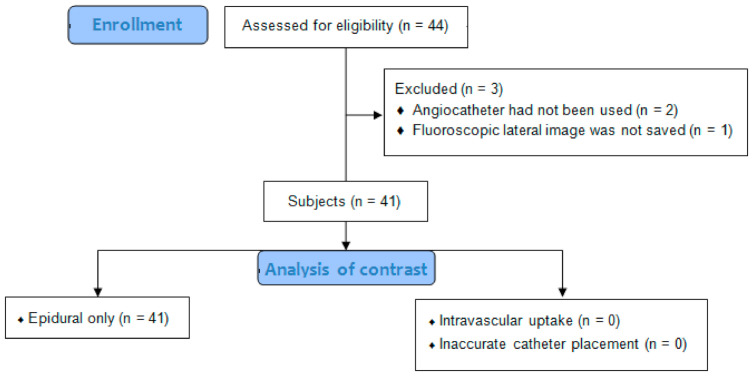
Flow diagram of the study.

**Figure 3 medicina-57-00318-f003:**
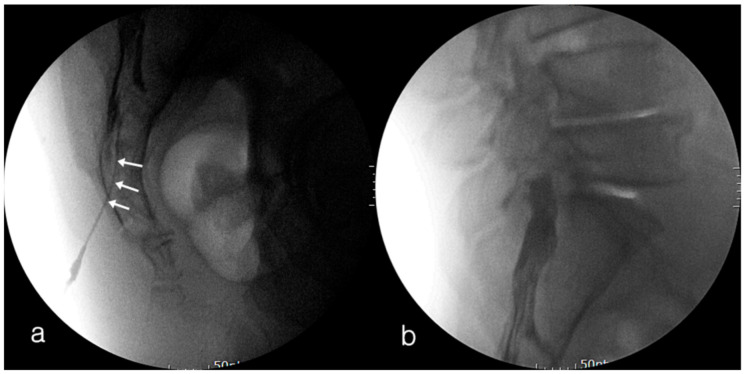
Fluoroscopic lateral images showing the contrast spread pattern through the 18-gauge, 1.88-inch catheter into the caudal epidural space. The outer catheter located within the sacral epidural space (**a**). The cephalic longitudinal spread of the contrast in caudal epidural space (**b**). Arrows indicate the outer catheter.

**Table 1 medicina-57-00318-t001:** Demographic data of 41 cases.

		Overall (*n* = 41)	Frequency (%)
Diagnosis	Spinal stenosis	24	58.5
	Herniated disc	17	41.5
Lesion level	L4-L5	16	39.0
	L5-S1	16	39.0
	Multi-level	9	22.0
Duration of pain	<3 moths	6	14.6
	3–12 months	11	26.9
	>12 months	24	58.5
Radicular pain	Presence	30	73.2
	Absence	11	26.8

All data are presented as the number or percentage of patients. Multi-level: L4-S1(7); L3-L5(2).

**Table 2 medicina-57-00318-t002:** Observational data of 41 cases.

		Overall (*n* = 41)	Frequency (%)
Intravascular signal	Epidural only	41	100.0
	Intravascular uptake	0	0
Level of ventral spread	L5	12	29.3
	S1	23	56.1
	<S1	6	14.6

All data are presented as the number or percentage of patients.

## Data Availability

The data presented in this study are available on request from the corresponding author.

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
