# Peer review of "The Influence of Ultrasound-Guided Technique Using a Catheter-Over-Needle on the Incidence of Accidental Intravascular Injection during Caudal Block-A Retrospective Case Series Study"

_medicina, 2021, doi:10.3390/medicina57040318_

Round 1

Reviewer 1 Report

The author highlights two main points of caudal technique: US guidance and catheter over needle. Both techniques are widely adopted by many institutions and are important techniques in interventional pain management. The authors suggest that it is the combination of the two techniques together that may lead to improved overall success of the procedure based on the avoidance of intravascular injections. However, this is somewhat unclear in the manuscript. 

In the cited reference, Doo et al. already demonstrate that US guidance with careful attention to depth already has a 0% intravascular rate. At the same time, it is mentioned in both the introduction and discussion that US is unable to detect intravascular injection. There is an important distinction that US may avoid intravascular injection but it does not detect when it does happen which is an advantage of fluoroscopy. There is no clear evidence that a catheter over needle technique improves on that further. The author concludes that it is superior for avoiding inadvertent intravascular injection but historically this does not seem to be the case nor does the author provide a control group to show superiority in the manuscript.

The authors can improve this manuscript by better linking the findings to a better supported conclusion. 

Some major areas of improvement include:

Abstract:

Line 23-34: it is unclear what the author means by "easy to control" Although routinely done, the author's data does not suggest that it can be used as standard of care technique over fluoroscopy. 

Conclusion:

Line 154-155: "the introduced outer catheter characteristics could have contributed to intravascular placement" perhaps the author intended for it to be the opposite 

Author Response

Thank you for your kind comments. We got a lot of help from you in revising my manuscript. We considered your opinions as much as possible. We appreciate your reviewing and look forward to receiving further response.

The author highlights two main points of caudal technique: US guidance and catheter over needle. Both techniques are widely adopted by many institutions and are important techniques in interventional pain management. The authors suggest that it is the combination of the two techniques together that may lead to improved overall success of the procedure based on the avoidance of intravascular injections. However, this is somewhat unclear in the manuscript.

In the cited reference, Doo et al. already demonstrate that US guidance with careful attention to depth already has a 0% intravascular rate. At the same time, it is mentioned in both the introduction and discussion that US is unable to detect intravascular injection. There is an important distinction that US may avoid intravascular injection but it does not detect when it does happen which is an advantage of fluoroscopy. There is no clear evidence that a catheter over needle technique improves on that further.

Response: We agree with your opinion. Doo et al. demonstrated the incidence of intravascular injections of the new caudal technique was 0 %. However, US-guided conventional technique using a catheter-over-needle induced successful block without inadvertent intravascular injection. Our results suggest that this technique is a reliable solution for conventional caudal block. We think that further controlled studies that compare the efficacy between US-guided conventional caudal block with catheter-over-needle and alternative technique are needed. And, US is unable to detect intravascular injection. However, this technique induced successful block without intravascular injection. We suggest that it can reduce the occurrence of intravascular injection itself. We added this finding in the limitation section. Lines 197-203.

The author concludes that it is superior for avoiding inadvertent intravascular injection but historically this does not seem to be the case nor does the author provide a control group to show superiority in the manuscript.

Response: We agree with your opinion. As we stated in discussion, there was no control group in this study. However, the intravascular incidence appears to compare well to published rates. And, we modified title and abstract.

The authors can improve this manuscript by better linking the findings to a better supported conclusion.

Some major areas of improvement include:

Abstract:

Line 23-34: it is unclear what the author means by "easy to control" Although routinely done, the author's data does not suggest that it can be used as standard of care technique over fluoroscopy.

Response: We agree with your opinion. We modified this point. Line 23-24.

Conclusion:

Line 154-155: "the introduced outer catheter characteristics could have contributed to intravascular placement" perhaps the author intended for it to be the opposite

Response: We agree with your opinion. We described it correctly. Line 165.

Reviewer 2 Report

Author reported the experience of caudal block using a catheter-over-needle under ultrasound guidance and evaluated the incidence of inadvertent intravascular injection in a retrospective manner. The author suggested the technique might be useful to avoid the inadvertent intravascular injection during the caudal block.

Broad comments

The manuscript was described concisely. As the authors stated in discussion, there was no control group in this study. In other words, this report can be considered as a case series. Discussion is a little bit redundant.

Specific comments

Title: As I mentioned above, it might be better to consider this report as a case series.

Methods:

P2, L45: During the study period, did author perform all the ultrasound-guided caudal block with a catheter-over-needle? It seems to be better and more precise to describe the inclusion criteria as “ultrasound-guided caudal block using a catheter-over-needle”.

P2, L73: What was the definition to identify the epidural space or to confirm that the needle tip was within the epidural space. What did author do if the needle tip was identified outside the epidural space or in the vascular structure when 1.0 mL of contrast media was injected?

P2, L75: I think it better to provide the information on the drugs administered after confirming the needle tip position for the treatment.

P2, L77: Please provide the definitions of epidural pattern, intravascular pattern, and intravascular-plus-epidural pattern of the fluoroscopic views. If the contrast media was directly injected in the vascular structure, it might disappear from the image within a few seconds and it might not be recorded appropriately on the pictures, which resulted in overlooking the intravascular injection. This might be another limitation of this study.

Results

Adverse events during procedure should be mentioned in this section.

Discussion:

In the previous report by Doo et al. (reference #11), the incidence of inadvertent intravascular injection and the rate of successful procedure during an ultrasound-guided caudal block using a conventional technique was reported to be 24% and 68%, respectively. However, when the needle advancement was stopped immediately after penetrating the sacrococcygeal ligament (new method), the incidence of inadvertent intravascular injection was 0% and the rate of successful procedure was 96%. The needle tip position in the new method was almost identical to the present report. This fact should be stated and discussed properly in this section. Furthermore, it also seems to be better to compare the new method described in the report by Doo and the catheter-over-needle technique described in this study in this section. What was the advantages of the catheter-over-needle technique?

Conclusion:

Although the author stated that the catheter-over-needle caudal block technique was recommended especially when the fluoroscopy was not available, it’s not based on the findings obtained from this study.

Author Response

Thank you for your kind comments. We got a lot of help from you in revising my manuscript. We considered your opinions as much as possible. We appreciate your reviewing and look forward to receiving further response.

Broad comments

The manuscript was described concisely. As the authors stated in discussion, there was no control group in this study. In other words, this report can be considered as a case series. Discussion is a little bit redundant.

Specific comments

Title: As I mentioned above, it might be better to consider this report as a case series.

Response: We agree with your opinion. We modified title. Line 4.

Methods:

P2, L45: During the study period, did author perform all the ultrasound-guided caudal block with a catheter-over-needle? It seems to be better and more precise to describe the inclusion criteria as “ultrasound-guided caudal block using a catheter-over-needle”.

Response: We agree with your opinion. We modified this point. Line 49.

P2, L73: What was the definition to identify the epidural space or to confirm that the needle tip was within the epidural space. What did author do if the needle tip was identified outside the epidural space or in the vascular structure when 1.0 mL of contrast media was injected?

Response: Optimal placement was defined as an epidural space filling on AP view and a placement of needle within the sacral epidural space. And, we did not find intravascular injection or extra-epidural spread in our cases when 1.0 mL of contrast media was injected. Lines 75-76.

P2, L75: I think it better to provide the information on the drugs administered after confirming the needle tip position for the treatment.

Response: We provided the information of the medications. Line 78-80.

P2, L77: Please provide the definitions of epidural pattern, intravascular pattern, and intravascular-plus-epidural pattern of the fluoroscopic views. If the contrast media was directly injected in the vascular structure, it might disappear from the image within a few seconds and it might not be recorded appropriately on the pictures, which resulted in overlooking the intravascular injection. This might be another limitation of this study.

Response: We described definitions, and we added a detection limitation. Lines 84-87. Lines 189-190.

Results

Adverse events during procedure should be mentioned in this section.

Response: We mentioned adverse events. Line 106.

Discussion:

In the previous report by Doo et al. (reference #11), the incidence of inadvertent intravascular injection and the rate of successful procedure during an ultrasound-guided caudal block using a conventional technique was reported to be 24% and 68%, respectively. However, when the needle advancement was stopped immediately after penetrating the sacrococcygeal ligament (new method), the incidence of inadvertent intravascular injection was 0% and the rate of successful procedure was 96%. The needle tip position in the new method was almost identical to the present report. This fact should be stated and discussed properly in this section. Furthermore, it also seems to be better to compare the new method described in the report by Doo and the catheter-over-needle technique described in this study in this section. What was the advantages of the catheter-over-needle technique?

Response: Thank you for your opinion. Doo et al. have mentioned an alternative method that states that the needle passes through the sacrococcygeal ligament but does not advance into the sacral canal as it is difficult to advance without touching the vulnerable region. We added this finding in the limitation section. We think further controlled studies that compare the efficacy between US-guided conventional caudal block with catheter-over-needle and alternative technique are needed.

Lines 197-203.

Conclusion:

Although the author stated that the catheter-over-needle caudal block technique was recommended especially when the fluoroscopy was not available, it’s not based on the findings obtained from this study.

Response: Thank you for your opinion. We revised the conclusion section. Lines 210-213.

Round 2

Reviewer 1 Report

Overall, the manuscript is improved but could use review for English language. I have not included all the areas, just one below. In addition, there are two points that should be addressed outlined below. 

Abstract- 

Line 24- an example of the minor English corrections please consider "this technique is a reliable option"

Line 25- I'm not sure that it is superior and therefore preferred as this study isn't a comparative study. Rather it is shown that this is a potential option. There are other things to consider when choosing technique. 

Introduction-

Line 34- The limitations of US's inability to detect intravascular injection is always present. By aiming to evaluate the influence of US in this technique doesn't change that. In my opinion the aim of this study is to say that US and catheter caudal injections don't result in intravascular injections which would mean that the inability to detect intravascular injections are a moot point. This is an assumption and not clearly stated in the introduction. The introduction the way it is written seems like the aim is to detect intravascular injection by US. 

Author Response

Thank you for your kind comments. We got a lot of help from you in revising my manuscript.

Our manuscript was edited by the professional English editors. We will correct any additional errors.

Overall, the manuscript is improved but could use review for English language. I have not included all the areas, just one below. In addition, there are two points that should be addressed outlined below. 

Abstract- 

Line 24- an example of the minor English corrections please consider "this technique is a reliable option"

Response: We revised it. Line 24.

Line 25- I'm not sure that it is superior and therefore preferred as this study isn't a comparative study. Rather it is shown that this is a potential option. There are other things to consider when choosing technique. 

Response: We deleted it. Line 24.

Introduction-

Line 34- The limitations of US's inability to detect intravascular injection is always present. By aiming to evaluate the influence of US in this technique doesn't change that. In my opinion the aim of this study is to say that US and catheter caudal injections don't result in intravascular injections which would mean that the inability to detect intravascular injections are a moot point. This is an assumption and not clearly stated in the introduction. The introduction the way it is written seems like the aim is to detect intravascular injection by US. 

Response: We mentioned it. Line 40-42.

Reviewer 2 Report

Broad comments

Author responded almost properly to my request. I added some minor comments to be considered for revision. In addition, please forgive my adding new requests that was not appeared in the previous review.

Specific comments

Abstract:

L2: A space before hyphen is not required.

Materials and Methods:

L56: “excluded from the study” not “excluded the study”.

L68: I think the following expression may be better to understand “...sacrococcygeal ligament visualized as hyperechoic band like structure in the ultrasound image.”

L72-74: To be precise, the expression of this sentence might be incorrect. Judging from the left side image of figure 3, it may be more precise to describe that the hollow flexible tube was advanced until the green hub reached to the skin surface. In other words, some part of the flexible tube was not within the epidural space but within the subcutaneous tissue.

L75: I guess authors confirm proper catheter placement instead of needle. Please make sure the details. Likewise, “needle” may be replaced by “catheter” for more precise expression in line 78.

Results

L104: It’s not necessary to calculate the average age to two places of decimals. The first place after the decimal point is enough.

L104-106: “We used contrast media fluoroscopy to confirm ...” This sentence is placed in the methods section. If my interpretation was correct, this has already been described in the methods (L75-77). Therefore, this sentence can be removed.

Discussion:

L149: Please delete “of” after because.

L172: I think biocompatibility of the catheter material has little importance for the inadvertent intravascular placement of the catheter. If catheter was placed in the epidural space for longer period, it may matter. In this occasion, flexibility of the catheter might be more important in contrast to the solid metal needle.

L178: I think it better to add “, respectively” after “...the vessel wall and the intravascular space”.

L217-223: Discussion related to the previous report by Doo et al. seems to be very important part of this study. I think that needle advancement just beyond the sacrococcygeal ligament is one of the most important technical aspect of the novel technique described in this report. Therefore, this topic should not be treated in the paragraph described the limitation of the study. The advantage of the new method compared to the report by Doo et al. is to advance only the flexible catheter beyond the sacrococcygeal ligament. This might be affect the ventral spread of contrast media because injection point became more cephalad to the sacrococcygeal ligament. Therefore, it might be better to discuss this topic in the fifth paragraph (L185-192).

Conclusion:

“a reliable solution for conventional caudal block” is an ambiguous expression. Let me provide the following example for one of the sentences for conclusion. “US-guided caudal block using catheter-over-needle can avoid inadvertent intravascular injection that has been a problem during caudal block using conventional technique.”

Tables:

I think it better to divide tables into two. One table for the demographic data of the patients, including gender, age, diagnosis, lesion level, duration of pain, existence of radicular pain. In another table, observational data including the contrast media spread pattern and level of ventral spread of the contrast media.

Figure:

In the legend for figure 2, Author use “the US-guided no-touch technique”. I guess this mean that metallic guide needle doesn’t touch the bone. If author use this phrase here, it may be better to use the same expression in the main text. Likewise, “stylet” appears only once in this legend. It may be better to use “the metallic guide needle” instead of stylet. The block arrow that indicates the sacrococcygeal ligament is difficult to notice. Why don’t you use white arrow heads? Also, it may be better to mention the ligament in the legend. For example, “The outer catheter is advanced beyond the sacrococcygeal ligament (white arrow heads) into the caudal epidural space (white dotted arrows).” It’s better to indicate the orientation of the image (cephalad/caudad).

In the legend of figure 3, it should be clearly mentioned that these images are the lateral view. Please mention the difference of the left and right images.

Author Response

Thank you for your kind comments. We got a lot of help from you in revising my manuscript. We considered your opinions as much as possible.

Broad comments

Author responded almost properly to my request. I added some minor comments to be considered for revision. In addition, please forgive my adding new requests that was not appeared in the previous review.

Specific comments

Abstract:

L2: A space before hyphen is not required.

Response: I am sorry, we could not find “a space before hyphen” in Line 2.

We ask you to recheck, please.

Materials and Methods:

L56: “excluded from the study” not “excluded the study”.

Response: We modified it. Line 56.

L68: I think the following expression may be better to understand “...sacrococcygeal ligament visualized as hyperechoic band like structure in the ultrasound image.”

Response: We modified it. Line 68.

L72-74: To be precise, the expression of this sentence might be incorrect. Judging from the left side image of figure 3, it may be more precise to describe that the hollow flexible tube was advanced until the green hub reached to the skin surface. In other words, some part of the flexible tube was not within the epidural space but within the subcutaneous tissue.

Response: We modified it. Line 73-74

L75: I guess authors confirm proper catheter placement instead of needle. Please make sure the details. Likewise, “needle” may be replaced by “catheter” for more precise expression in line 78.

Response: We modified it. Line 75.

Results

L104: It’s not necessary to calculate the average age to two places of decimals. The first place after the decimal point is enough.

Response: We modified it. Line 103.

L104-106: “We used contrast media fluoroscopy to confirm ...” This sentence is placed in the methods section. If my interpretation was correct, this has already been described in the methods (L75-77). Therefore, this sentence can be removed.

Response: Thank you for your detailed review. We removed it.

Discussion:

L149: Please delete “of” after because.

Response: We modified it. Line 152.

L172: I think biocompatibility of the catheter material has little importance for the inadvertent intravascular placement of the catheter. If catheter was placed in the epidural space for longer period, it may matter. In this occasion, flexibility of the catheter might be more important in contrast to the solid metal needle.

Response: We modified it. Line 173-174

L178: I think it better to add “, respectively” after “...the vessel wall and the intravascular space”.

Response: We added it. Line 180.

L217-223: Discussion related to the previous report by Doo et al. seems to be very important part of this study. I think that needle advancement just beyond the sacrococcygeal ligament is one of the most important technical aspect of the novel technique described in this report. Therefore, this topic should not be treated in the paragraph described the limitation of the study. The advantage of the new method compared to the report by Doo et al. is to advance only the flexible catheter beyond the sacrococcygeal ligament. This might be affect the ventral spread of contrast media because injection point became more cephalad to the sacrococcygeal ligament. Therefore, it might be better to discuss this topic in the fifth paragraph (L185-192).

Response: We modified it. This content was moved to the fifth paragraph. Line 187-193

Conclusion:

“a reliable solution for conventional caudal block” is an ambiguous expression. Let me provide the following example for one of the sentences for conclusion. “US-guided caudal block using catheter-over-needle can avoid inadvertent intravascular injection that has been a problem during caudal block using conventional technique.”

Response: Thank you for your suggestion. We modified it. Line 217-219

Tables:

I think it better to divide tables into two. One table for the demographic data of the patients, including gender, age, diagnosis, lesion level, duration of pain, existence of radicular pain. In another table, observational data including the contrast media spread pattern and level of ventral spread of the contrast media.

Response: We divided tables into two. Table 1,2. Page 3,4

Figure:

In the legend for figure 2, Author use “the US-guided no-touch technique”. I guess this mean that metallic guide needle doesn’t touch the bone. If author use this phrase here, it may be better to use the same expression in the main text. Likewise, “stylet” appears only once in this legend. It may be better to use “the metallic guide needle” instead of stylet. The block arrow that indicates the sacrococcygeal ligament is difficult to notice. Why don’t you use white arrow heads? Also, it may be better to mention the ligament in the legend. For example, “The outer catheter is advanced beyond the sacrococcygeal ligament (white arrow heads) into the caudal epidural space (white dotted arrows).” It’s better to indicate the orientation of the image (cephalad/caudad).

Response: Thank you for your opinion. We deleted and modified those.

And, we changed figure 2 and legend. Page 5.

In the legend of figure 3, it should be clearly mentioned that these images are the lateral view. Please mention the difference of the left and right images.

Response: We modified figure 3 and legend. Page 5.
